# Measuring heterogeneity in normative models as the effective number of deviation patterns

**Abraham Nunes** [1,¤]*, **Thomas Trappenberg** [2], **Martin Alda** [1]

**1** Department of Psychiatry, Dalhousie University, Halifax, Nova Scotia, Canada, **2** Faculty of Computer Science, Dalhousie University, Halifax, Nova Scotia, Canada

¤ Current address: QEII Health Sciences Centre, Halifax, Nova Scotia, Canada
* nunes@dal.ca

**Data Availability Statement:** All relevant data are within the manuscript and its Supporting information files.

**Funding:** Ruth Wagner Memorial Fund (AN). The funders had no role in study design, data collection

## Abstract

Normative modeling is an increasingly popular method for characterizing the ways in which clinical cohorts deviate from a reference population, with respect to one or more biological features. In this paper, we extend the normative modeling framework with an approach for measuring the *amount* of heterogeneity in a cohort. This heterogeneity measure is based on the *Representational Rényi Heterogeneity* method, which generalizes diversity measurement paradigms used across multiple scientific disciplines. We propose that heterogeneity in the normative modeling setting can be measured as *the effective number of deviation patterns*; that is, the effective number of coherent patterns by which a sample of data differ from a distribution of normative variation. We show that *lower* effective number of deviation patterns is associated with the presence of systematic differences from a (non-degenerate) normative distribution. This finding is shown to be consistent across (A) application of a Gaussian process model to synthetic and real-world neuroimaging data, and (B) application of a variational autoencoder to well-understood database of handwritten images.

## Introduction

Psychiatric disorders are defined by their clinical presentations, which means that several different biological abnormalities could result in what we erroneously call a single condition. This has important consequences for biological and treatment studies that assume subjects form a single homogeneous population. Instead, in reality, a given psychiatric condition such as bipolar disorder might differ from normal population variation in heterogeneous ways. Normative modeling is a popular method for disentangling this heterogeneity in clinical cohorts [1–10]. This method involves learning a distribution of normal variation, with the assumption that clinically relevant phenotypes are identifiable by significant deviations from this normative range. Unfortunately, it does not measure the *amount* of heterogeneity in a cohort. To improve our understanding of psychiatric nosology, we must aim to understand the factors that cause biological and other forms of heterogeneity in our diagnostic system, as

and analysis, decision to publish, or preparation of
the manuscript.

**Competing interests:** The authors have declared
that no competing interests exist.

well as the consequences of that heterogeneity on research and clinical practice. However, to
study heterogeneity rigorously, we must be able to measure it rigorously. To date, there is no
proposed method to rigorously measure heterogeneity in normative models.

Heterogeneity measurement has been studied for more than a century [11], but most indices, such as variance and entropies [12–14], have inconsistent units and can scale counterintuitively [15, 16]. Conversely, ecologists and others have adopted the *Rényi heterogeneity* family
of indices as a "true diversity" index [17–20]. The Rényi heterogeneity measures a system's
effective number of configurations (*numbers equivalent* [21]). This measure scales linearly and
generalizes most commonly used diversity indices [15, 20, 22–24].

We therefore introduce a Rényi heterogeneity measurement for normative modeling studies, the *effective number of deviation patterns* (ENDevs), which estimates the number of distinct
ways in which a cohort deviates from the normative distribution. To our knowledge, this is the
first such explicitly defined heterogeneity measurement approach for the normative modeling
framework. We also demonstrate a useful property of ENDevs: that cohorts sampled from *outside* a non-trivial normative distribution will tend to have lower ENDevs than cohorts sampled
from the normative distribution proper.

## Background

### Normative modeling

Normative modeling involves four steps [1]. First, one defines the spaces of predictor
and response variables, $\mathcal{X} = \mathbb{R}^K$ and $\mathcal{Y} = \mathbb{R}^M$, respectively. A dataset
$\mathcal{D}_o = \{(\mathbf{x}_i, \mathbf{y}_i) : i \in \{1, 2, \ldots, N\}\}$ is collected for a "normative" cohort comprising predictor $\mathbf{x} \in \mathcal{X}$ and response variables $\mathbf{y} \in \mathcal{Y}$. Often, the predictor variables are clinical covariates
such as behavioural traits, sex, and age, with the response variable being some biological
measurement such as the volume of some brain region on structural neuroimaging.

The second and third steps involve learning a model of the mapping $\mathcal{X} \rightarrow \mathcal{Y}$ using the
normative cohort's data $\mathcal{D}_o$, with out-of-sample model criticism. Ideally, the model should
map predictors onto a space of probability distributions on the response variable, that is
$\mathbf{g} : \mathcal{X} \rightarrow \mathcal{P}(\mathcal{Y})$ [1].

Finally, one collects dataset $\mathcal{D}' = \{(\mathbf{x}'_i, \mathbf{y}'_i) : i \in \{1, 2, \ldots, N'\}\}$ from a target cohort of $N'$
subjects, such as clinical patients. One then computes the degree to which $\mathbf{Y}'$ deviates from the
predictions given by $\mathbf{g}(\mathbf{X}')$. A common approach to quantifying this deviation when $\mathbf{g}$ is a
Gaussian process regression model is a Z-score that we henceforth call the "standardized deviance." For each subject $i \in \{1, 2, \ldots, N'\}$ and response variable $j \in \{1, 2, \ldots, M\}$ the standardized deviance is

$$Z_{ij} = \frac{y'_{ij} - \mu_j(\mathbf{x}'_i)}{\sqrt{\sigma_j^2(\mathbf{x}'_i) + \eta_j^2}} \tag{1}$$

where $\mu_j(\mathbf{x}'_i)$ and $\sigma_j^2(\mathbf{x}'_i)$ are the expected value and predicted variance, respectively, given
predictors $\mathbf{x}'_i$, and $\eta_j^2$ is the variance in the response variable learned by the normative
model. Statistics summarizing deviance at the subject level can then be computed in various
ways, including summation or average deviance, thresholding of the Z-score [5], or by
application of extreme value statistics to model largest deviations [25]. However, there are
no existing approaches for quantifying the absolute amount of heterogeneity using normative modeling.

## Measuring heterogeneity

Heterogeneity is the degree to which a system deviates from perfect conformity, and is measured as the effective size of the system's event space using the Rényi heterogeneity indices [18, 19, 22, 26]. Given a system with probability distribution $\mathbf{p} = (p_i)_{i=1,2,\ldots,n}$ over $n$ categorical states, the Rényi heterogeneity of order $q \geq 0$ is defined as follows:

$$\Pi_q(\mathbf{p}) = \begin{cases} \sum_{i=1}^{n} \mathbb{1}[p_i > 0] & q = 0 \\ \exp\left\{-\sum_{i=1}^{n} p_i \log p_i\right\} & q = 1 \\ (\max_i p_i)^{-1} & q = \infty \\ \left(\sum_{i=1}^{n} p_i^q\right)^{\frac{1}{1-q}} & \text{Otherwise} \end{cases}, \tag{2}$$

where $\mathbb{1}[x]$ is an indicator function. At $q \notin \{0, 1, \infty\}$, the continuous analogue for some random variable $X$ with event space $\mathcal{X}$ and probability density function $f$ is

$$\Pi_q(X) = \left(\int_{\mathcal{X}} f(\mathbf{x}) d\mathbf{x}\right)^{\frac{1}{1-q}}, \tag{3}$$

which corresponds to the effective hypervolume of $\mathcal{X}$. The parameter $q$ defines the indices' insensitivity to rare configurations (S1 Fig). A critical property of Rényi heterogeneity is satisfaction of *the replication principle* [15], which guarantees that Eqs 2 and 3 scale linearly with the system's number of distinct configurations [22, 24]. In other words, unlike variance and entropies, a 50% increase in Rényi heterogeneity corresponds to a 50% increase of variation in the system (illustrated graphically in S2 Fig, but for a proof of the replication principle see Nunes et al. [22]).

## The effective number of deviation patterns

Let $\mathbf{Z}^* = |\mathbf{Z}| = (\mathbf{z}_i^*)_{i=1,2,\ldots,N}$ be the $N \times M$ absolute standardized deviance matrix, where $\mathbf{z}_i^*$ is the absolute standardized deviance for subject $i \in \{1, 2, \ldots, N\}$. We set the bottom $100(1-c)\%$ of values in $\mathbf{z}_i^*$ to 0 and rescale the remainder such that they sum to one, thereby generating a probability distribution over the $c$-deviant features (i.e. those features with the top $100c\%$ of standardized deviance values):

$$\begin{aligned} \tilde{\psi}(\mathbf{z}_i^*, c) &= \left\{\mathbb{1}[\hat{S}(z_{ij}^*) < c] z_{ij}^*\right\}_{j=1,2,\ldots,M} \\ \psi(\mathbf{z}_i^*, c) &= \left\{\frac{\tilde{\psi}_j(\mathbf{z}_i^*, c)}{\sum_{k=1}^{M} \tilde{\psi}_k(\mathbf{z}_i^*, c)}\right\}_{j=1,2,\ldots,M} \end{aligned} \tag{4}$$

where $\hat{S} : \mathbb{R}_+ \to [0, 1]$ is the empirical survival function over elements of $\mathbf{z}_i^*$. However, note that it is not necessary to use the absolute value of $\mathbf{Z}$ (one could use the $100c\%$ highest or lowest values if desired). By Eq 2, the effective number of $c$-deviant features for subject $i$ is thus

$$\Pi_q(\mathbf{z}_i^*, c) = \left(\sum_{j=1}^{M} \psi_j^q(\mathbf{z}_i^*, c)\right)^{\frac{1}{1-q}}. \tag{5}$$

Since each subject is associated with his or her own pattern of $c$-deviant features, the effective number of deviation patterns (*ENDevs*) is essentially a measure of the effective number of distinct subjects in the sample with respect to the representation generated by the normative

model. The ENDevs is computed by solving the following decomposition for the $\Pi_q^\beta(\mathbf{Z}^*, c)$ term [20]:

$$\underbrace{\Pi_q^\gamma(\mathbf{Z}^*, c)}_{\substack{\text{Effective number of} \\ c-\text{deviant features in} \\ \text{pooled sample}}} = \underbrace{\Pi_q^\alpha(\mathbf{Z}^*, c)}_{\substack{\text{Effective number of} \\ c-\text{deviant features} \\ \text{per subject}}} \times \underbrace{\Pi_q^\beta(\mathbf{Z}^*, c)}_{\substack{\text{Effective number} \\ \text{of subject}}} . \tag{6}$$

Eq 6 is interpreted as follows: the effective number of deviation patterns $\Pi_q^\beta(\mathbf{Z}^*, c)$ is equal to the *overall effective number of c-deviant features* in the pooled sample, $\Pi_q^\gamma(\mathbf{Z}^*, c)$, divided by the *effective number of c-deviant features per subject*, $\Pi_q^\alpha(\mathbf{Z}^*, c)$. This yields the effective number of subjects. However, since we are representing each subject by his or her pattern of $c$-deviant features, we use the more precise description of ENDevs.

We first compute the overall effective number of $c$-deviant features, $\Pi_q^\gamma(\mathbf{Z}^*, c)$, also known as the $\gamma$-heterogeneity in ecology,

$$\Pi_q^\gamma(\mathbf{Z}^*, c) = \left(\sum_{j=1}^M \bar{\psi}_j^q(\mathbf{Z}^*, c)\right)^{\frac{1}{1-q}}. \tag{7}$$

where

$$\bar{\psi}(\mathbf{Z}^*, c) = \sum_{i=1}^N w_i \, \psi(\mathbf{z}_i^*, c) \tag{8}$$

is a probability distribution over the $c$-deviant features in the *pooled* sample (i.e. across all subjects), and where $0 < w_i < 1$ is a weight assigned to the deviation pattern of subject $i$, such that $\sum_{i=1}^N w_i = 1$. In all examples in the present study, we set $w_i = 1/N$ for all individual observations or subjects.

We then compute the $\alpha$-heterogeneity, $\Pi_q^\alpha(\mathbf{Z}^*, c)$, which is effective number of $c$-deviant features per deviation pattern (i.e. per subject) as follows:

$$\Pi_q^\alpha(\mathbf{Z}^*, c) = \left(\frac{\sum_{i=1}^N w_i^q \sum_{j=1}^M \psi_j^q(\mathbf{z}_i^*, c)}{\sum_{k=1}^N w_k^q}\right)^{\frac{1}{1-q}}. \tag{9}$$

The ratio of $\gamma$- and $\alpha$-heterogeneity yields $\beta$-heterogeneity, $\Pi_q^\beta(\mathbf{Z}^*, c)$, whose units are ENDevs:

$$\Pi_q^\beta(\mathbf{Z}^*, c) = \frac{\Pi_q^\gamma(\mathbf{Z}^*, c)}{\Pi_q^\alpha(\mathbf{Z}^*, c)}. \tag{10}$$

In terms of representational Rényi heterogeneity [22], Eqs 7–10 measure the effective number of distinct observations in a cohort with respect to the representation $\mathbf{Z}^*$ generated by the normative model.

## Methods

### Data availability and ethics statements

Data and code for all analyses are provided as supplemental materials. Neuroimaging data were anonymized from the *Autism Brain Imaging Data Exchange* database (ABIDE; https://fcon_1000.projects.nitrc.org/indi/abide/), and preprocessed by Haar et al. [27]

(http://dinshi.com/abide/). All ABIDE sites received local Institutional Review Board approval for data acquisition.

## Experiment using synthetic data

Our first experiment uses synthetic data from a simple system of $M = 30$ real-valued features, $\mathbf{y}_i$—which may be considered analogous to 30 regions of interest in a neuroimaging study—using the following function parameterized by five simulated "covariates" (Fig 1A–1E):

$$\mathbf{y}_i = (x_{i1} - 2) \sin \{x_{i2}\pi\mathbf{t}\} \cos \{x_{i3}\pi\mathbf{t}\} - \cos \{\pi(\mathbf{t} - x_{i4})\} + \tilde{\epsilon}_i, \tag{11}$$

where

$$\mathbf{t} = \left\{ \frac{(j-1)(u-l)}{M-1} + l : j \in \{1, 2, \dots, M\}, l = -2, u = 2 \right\} \tag{12}$$

$$\tilde{\epsilon}_i = (\tilde{\epsilon}_{ij})_{j=1,2,\dots,M}, \quad \tilde{\epsilon}_{ij} \sim \mathcal{N}(x_{i5}, 0.1) \tag{13}$$

and where $\mathbf{x}_i = (x_{ij})_{j=1,2,\dots,5}$ are the covariates sampled from an isotropic multivariate Gaussian. Specific model parameters are included in the reproducible supplementary notebooks. The generative model specified by Eqs 11–13 was selected in order to generate nonlinear patterns that remain easy to visualize, yet can result in non-trivial patterns of differences between simulated groups. That is, two groups simulated under these data will not simply differ along one feature dimension, but could show heterogeneous differences across multiple features.

We generated data from this system for $N = 50$ subjects in a "normative cohort" defined by a specific parameterization of isotropic multivariate Gaussian distribution over covariates (Fig 1F). The sample size was chosen to be large enough to allow for clear visualization of differences, while remaining small enough that the analyses could be reasonably reproduced by readers running typical personal computers. Interested readers may manipulate all parameters of this analysis in the Supplementary code (S1 File). A Gaussian process normative model was fit to these data (implemented in GPyTorch v.1.1 [28]) and evaluated for generalizability on an independently sampled normative cohort (Fig 1G).

We then sampled an independent clinical cohort, with $N_{Unaffected} = 50$ subjects from the normative distribution (i.e. "unaffected" subjects), and $N_{Affected} = 50$ sampled from an "affected" group characterized by different covariate distributions (Fig 2A and 2B). The trained normative model was then used to compute absolute values of standardized deviance (Eq 1) for each subject in the clinical cohort.

If a set of individuals forms a single coherent group based on their clinical condition, the extreme values should be found in similar features (i.e. subjects will have a similar *deviation pattern*). Conversely, the within-subject extreme values in the normative group will tend to be more randomly distributed across the feature space. To evaluate this quantitatively, we computed ENDevs using the procedure in Eqs 4–10. Means and 95% confidence intervals for ENDevs were estimated using bootstrap sampling at various extreme value thresholds ($0 < c \leq 1$), and compared between the affected and unaffected groups.

## Experiment using the MNIST dataset

Our first experiment used synthetic data to provide a simple illustration of how a "clinical" or "target" cohort will have a lower ENDevs than a sample of individuals drawn from the distribution of normative variation. Our second experiment seeks to evaluate whether this same phenomenon will occur with more complex and high-dimensonal data, such as natural images

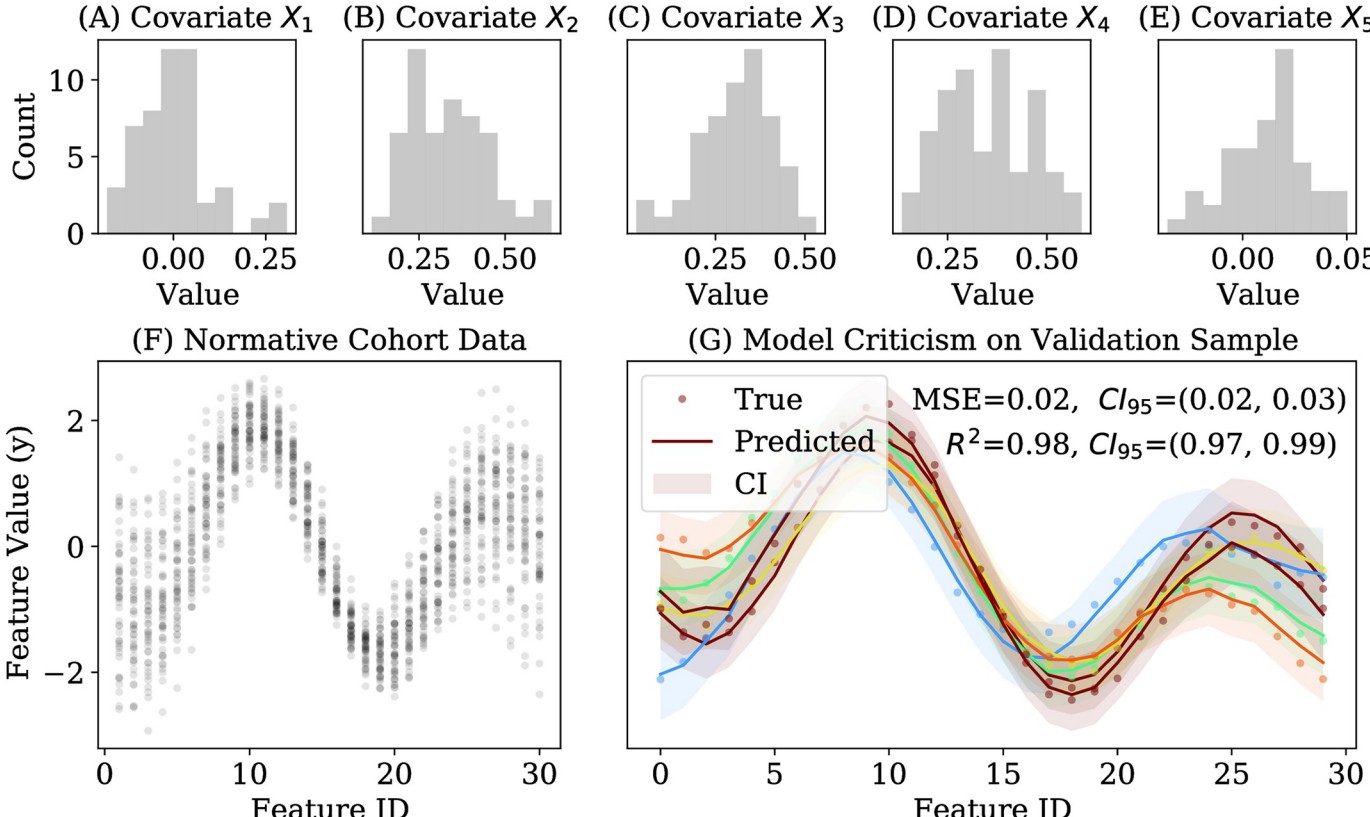

**Fig 1. Illustration of simulated data and normative model. Panels A-E**: Simulated distribution of five covariates. **Panel F**: Example of simulated data for $M = 30$ features and $N = 50$ subjects in a normative cohort. **Panel G**: Criticism of a normative model fit to the data in Panel A for 6 new subjects from the normative distribution (different color per subject).

of objects from different categorical classes. For such a dataset to be appropriate, we must be able to guarantee that there are meaningful differences between groups in the raw data, and that these differences can be easily visualized for illustrative purposes. We therefore use the MNIST dataset, which includes 70,000 images of handwritten digits, roughly evenly balanced over classes {0, 1, . . ., 9} [29]. The MNIST dataset is among the simplest non-trivial examples in which consistent deviation patterns exist between robustly-defined classes. That is, (A) digit classes are valid partitions, and (B) their use for written communication mandates the existence of consistent and circumscribed deviation patterns between classes.

We defined a "normative cohort" as a set of images belonging to one of the digit classes {0, 1, . . ., 9}. Normative variation in this cohort was modeled using a convolutional variational autoencoder (cVAE) [30, 31]. For some input data (here a 28-by-28 pixel image), a cVAE uses an encoder module to learn a compressed latent representation that carries sufficient information to reconstruct the input image via a decoder module. Both the encoder and decoders are convolutional neural networks. The objective function is a lower bound on the model evidence (the *evidence lower bound* or *ELBO*) whose maximization is equivalent to minimizing the Kullback-Leibler divergence between the approximate and unknown posteriors over latent representations. Further theoretical details can be found elsewhere [30, 31]. Our cVAE was implemented in PyTorch (v. 1.5 for Python v. 3.7) with an 8-dimensional latent space.

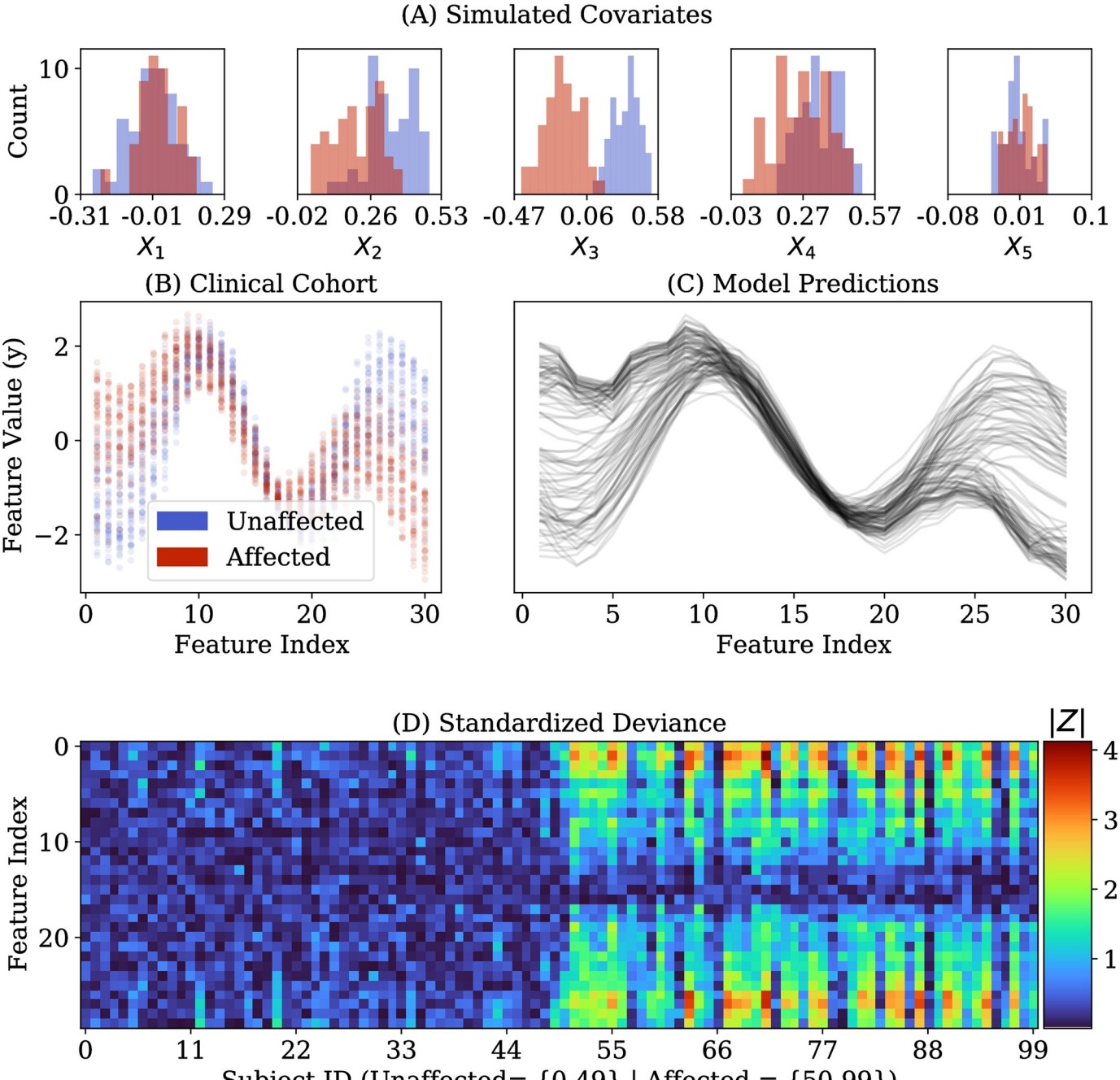

**Fig 2. Application of normative model to independent cohorts from synthetic normative and clinical distributions. Panel A**: Simulated covariates for $N' = 100$ subjects in a clinical cohort (50 "affected" and 50 "unaffected"). **Panel B**: Plot of the $M = 30$ response variables. **Panel C**: Predictions of a normative model trained on the normative cohort data. **Panel D**: Standardized deviance (Eq 1) of each subject, at each of the 30 features.

Normative variation of each digit class was modeled by fitting a cVAE to that digit's images under 10-fold cross-validation. Within each fold of this partitioning scheme, 90% of the respective digit's images were used to train the cVAE (200 epochs; S3 Fig). After training, we computed standardized deviance scores for the held out 10% of samples. When concatenated,

the scores of the held out partitions represent the standardized deviance for cohorts sampled *within* the distribution of normative variation. Standardized deviance was computed using Eq 1, although we substituted the variance (squared Euclidean distance) with the squared Fisher-Hotelling-Rao distance (which is the Riemannian metric length distance between multinomial distributions) [32, 33], since the output of the cVAE is a $28 \times 28$ matrix of Bernoulli parameters on pixel values.

After computing out-of-sample standardized deviation scores for the normative class, we trained a cVAE on *all samples* from the normative digit class. This model was used to estimate standardized deviation scores for other digit class' images, respectively. With respect to (normative) digit class $i$, all other digit classes can be considered the "clinical" or "affected" cohorts under the medical analogy.

Let $\mathbf{Z}_{ij}^*$ denote the $N_j \times 784$ matrix of pixel-wise absolute standardized deviation values computed for the $N_j$ samples in the "clinical" digit class $j \in \{0, 1, \ldots, 9\}$ under the model of variation in "normative" digit class $i \in \{0, 1, \ldots, 9\}$. Given $\mathbf{Z}_{ij}^*$ and an extreme value threshold $0 < c \leq 1$, the ENDevs in digit class $j$ under a normative model of class $i$ can be computed using Eqs 4–10 with 95% confidence intervals estimated using bootstrap sampling. We hypothesized that the ENDevs ($\beta$–heterogeneity; Eq 10) will be greatest when $i = j$; that is, when the "clinical" cohort is sampled from the normative distribution.

## Experiment using autism neuroimaging data

Finally, we apply our heterogeneity measure to structural neuroimaging data from ABIDE (preprocessed regions of interest from Haar et al. [27]). Since our purpose here is primarily methodological illustration, we included only those subjects for whom complete covariate data were provided for *site, sex, handedness*, and *age* (S1 Table; $N_{Control} = N_{Autism} = 199$).

Normative and clinical cohorts were defined as the control and autism-diagnosed subjects, respectively. The normative model was a Gaussian process implemented in the GPyTorch package (v.1.1) [28], with hyperparameter optimization done using gradient descent on the log model evidence. Generalizability was evaluated under 10-fold cross validation. Only features for which the model could explain an average $> 20\%$ of variance over folds were kept (132 features, listed in S2 Table). On each held out validation partition, we computed the standardized deviance using Eq 1, and concatenated the results to represent the deviation profiles for the normative cohort. The normative model was then trained on these 132 features using the entire normative sample to obtain standardized deviance scores for the clinical group.

We estimated ENDevs using boostrap sampling across multiple extreme value thresholds ($c \in \{0.01, 0.02, 0.03, 0.05, 0.07\}$, for the top $100c\%$ values of $|\mathbf{Z}|$, as well as the bottom $100c\%$ and top $100c\%$ of values of $\mathbf{Z}$) as per Eqs 4–10. Means and 95% confidence intervals were compared between the autism and control subsamples. We hypothesized that ENDevs for the control group would exceed those of the autism subsample.

Small samples may result in underestimation of the ENDevs, particularly if the number of ENDevs are large. In other words, one must observe many individuals if we are to count their large number of deviation patterns. To evaluate the degree to which our sample size is sufficient to capture ENDevs in control and autism groups, we plotted heterogeneity accumulation curves. These curves are constructed by re-estimating the ENDevs in each group using bootstrap subsampling at progressively larger sample sizes (we evaluated samples between $N = 5$ and $N = 199$ in increments of 10). If a clinical cohort deviates consistently from the normative distribution, one should observe the clinical group's accumulation curve plateau earlier than that of the normative cohort.

## Results

### Experiment using synthetic data

The normative model's predictions of the response variables for each subject in the clinical sample—given his or her covariates—are shown in Fig 2C. Fig 2D plots standardized deviance values for all subjects in the clinical cohort (i.e. unaffected and affected).

Fig 3A demonstrates the subject-wise probability distributions over the deviant features (i.e. $\psi(\mathbf{z}_i^*, c)$) for various extreme value thresholds $c \in \{1, 0.3, 0.05, 0.01\}$, stratified by group. Sparsity of the $c$-deviant feature distributions increases inversely proportional to $c$. Fig 3B shows that the distribution over $c$-deviant features in the pooled cohort (Eq 8) becomes more "peaked" with lower values of $c$. Together, Fig 3A and 3B suggest—and Fig 3C and 3D confirms—that the effective number of $c$-deviant features per subject ($\Pi_q^\alpha$; Eq 9) and overall ($\Pi_q^\gamma$; Eq 7) should increase with $c$.

Fig 3A also shows that the $c$-deviation patterns are consistent within the "affected" subsample. That is, the most $c$-deviant features tend to be the same across subjects, suggesting that they deviate from the normative distribution in similar ways. Conversely, the distribution of $c$-deviant features in the unaffected subsample spans a larger number of features, inconsistently

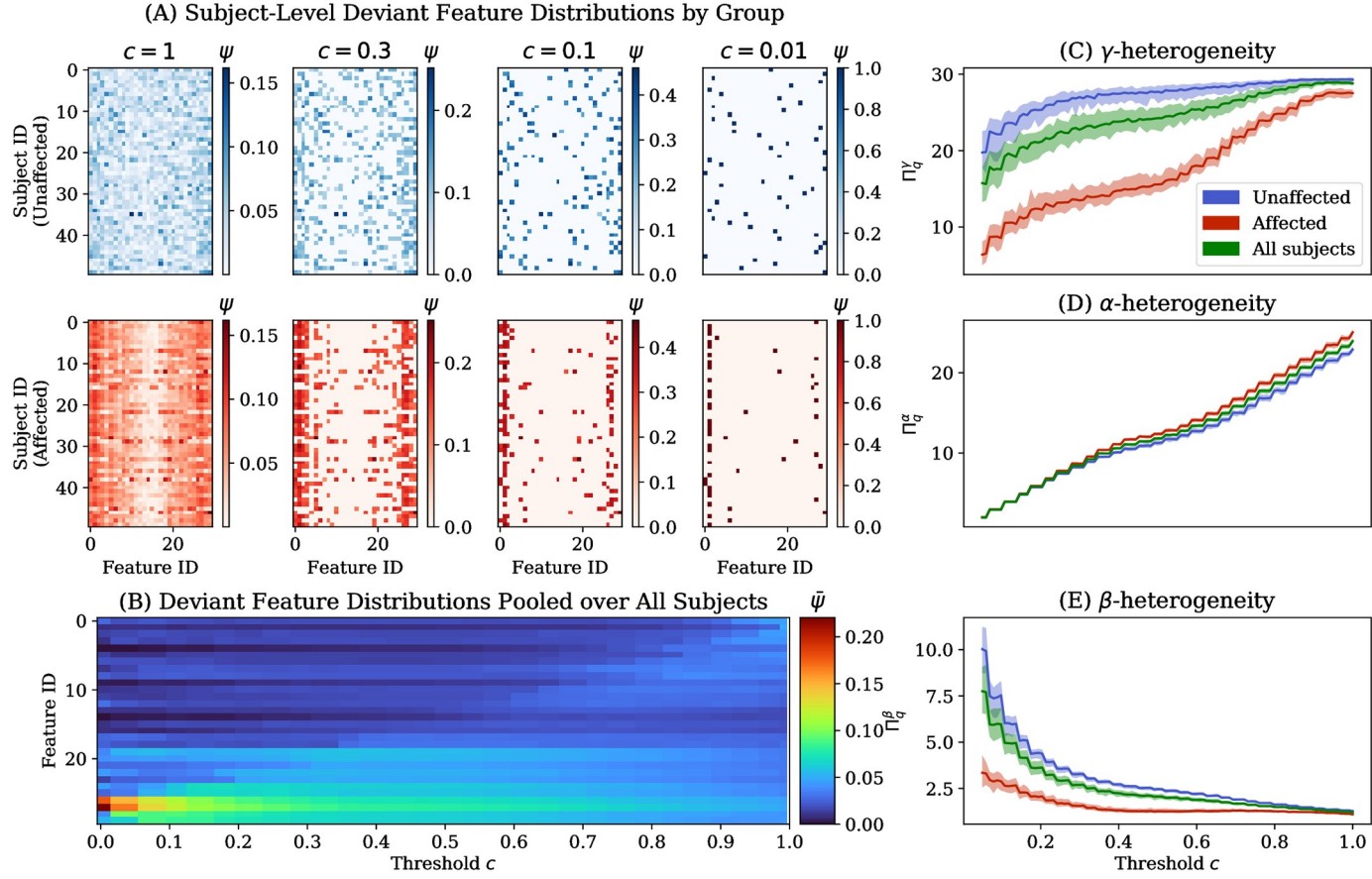

**Fig 3. Measurement of heterogeneity in the synthetic clinical cohort. Panel A**: Subject-level probability distributions over the most $c$-deviant (i.e. top $100c$%) features. Each row within a heatmap is a probability distribution computed using Eq 4, and columns show distributions across thresholds $c \in \{0.01, 0.1, 0.3, 1\}$. **Panel B**: Pooled distribution over $c$-deviant features ($\bar{\psi}(\mathbf{Z}^*, c)$; Eq 8), shown column-wise, across values of $c$ (on x-axis). **Panels C-E**: The $\gamma$ (Eq 7), $\alpha$ (Eq 9), and $\beta$ heterogeneity (Eq 10), respectively, in the unaffected (blue), affected (red), and pooled (green) cohorts. Solid lines are means, and shaded ribbons are 95% confidence intervals.

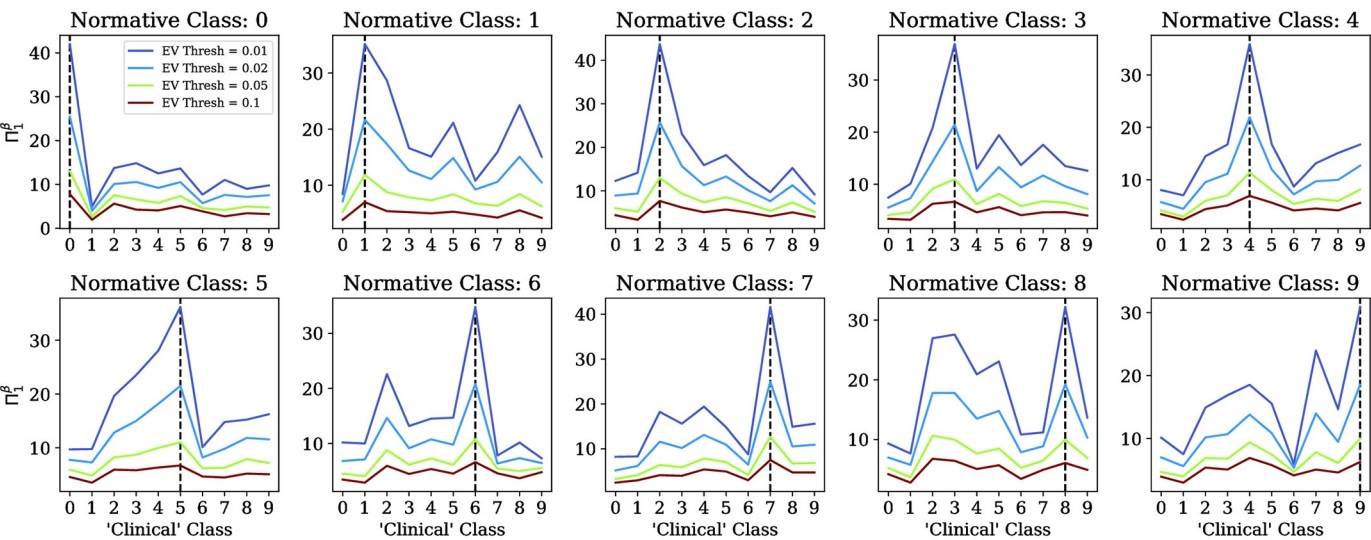

**Fig 4. Results of the MNIST experiment.** Each plot corresponds to a normative model (cVAE [30, 31]) trained on images from the "normative" digit class identified in the title. The x-axes index "clinical" or "target" digit classes. The y-axes plot the $\beta$-heterogeneity at $q = 1$ ($\Pi_1^\beta$, Eq 10), across thresholds $c \in \{0.01, 0.02, 0.05, 0.1\}$. Dashed vertical lines highlight maximal values of $\Pi_1^\beta$.

across subjects. Indeed, Fig 3E confirms that the affected group's ENDevs ($\Pi_q^\beta$; Eq 10) is lower than that of the unaffected subsample.

## Experiment using the MNIST dataset

Across multiple extreme value thresholds ($c \in \{0.01, 0.02, 0.05, 0.1\}$), Fig 4 confirms our hypothesis that the peak ENDevs occur when the clinical and normative cohorts are of the same digit class. The supplemental materials includee a plot of the cVAE training loss across each digit class (S3 Fig), and visualization of deviation profile distributions for each normative-clinical digit class combination (Fig 4).

## Experiment using autism neuroimaging data

There were no univariate differences between groups across covariates (S1 Table). Comparison of ENDevs between control and autism subgroups are shown in Fig 5 and Table 1 (across $c \in \{0.01, 0.02, 0.03, 0.05, 0.07\}$). At $c = 0.01$, ENDevs estimated for the autism group was lower than that of the control group for the top $100c\%$ absolute standardized deviances (41.1 95% CI [38.8,43.4] vs. 46.4 [44.4,48.4]) and for the top $100c\%$ standardized deviances (39.9 [37.8,42.1] vs. 46.8 [43.6,49.4]), albeit only slightly. There was no difference in ENDevs between autism and control samples computed with the bottom $100c\%$ standardized deviance values (41.7 [38.0,44.2] vs. 43.9 [40.8, 46.0]). As with our experiments on synthetic and MNIST data (Figs 3 and 4, respectively), differences in ENDevs between groups were attenuated as the extreme value threshold $c$ was increased.

Heterogeneity accumulation curves in Fig 5 show that (A) ENDevs in the autism group never exceeds that of the control group, (B) ENDevs in the autism group begin to plateau earlier than ENDevs in the control group, and (C) neither group achieves a consistent plateau value for ENDevs, suggesting that further sampling will likely discover further novel variation.

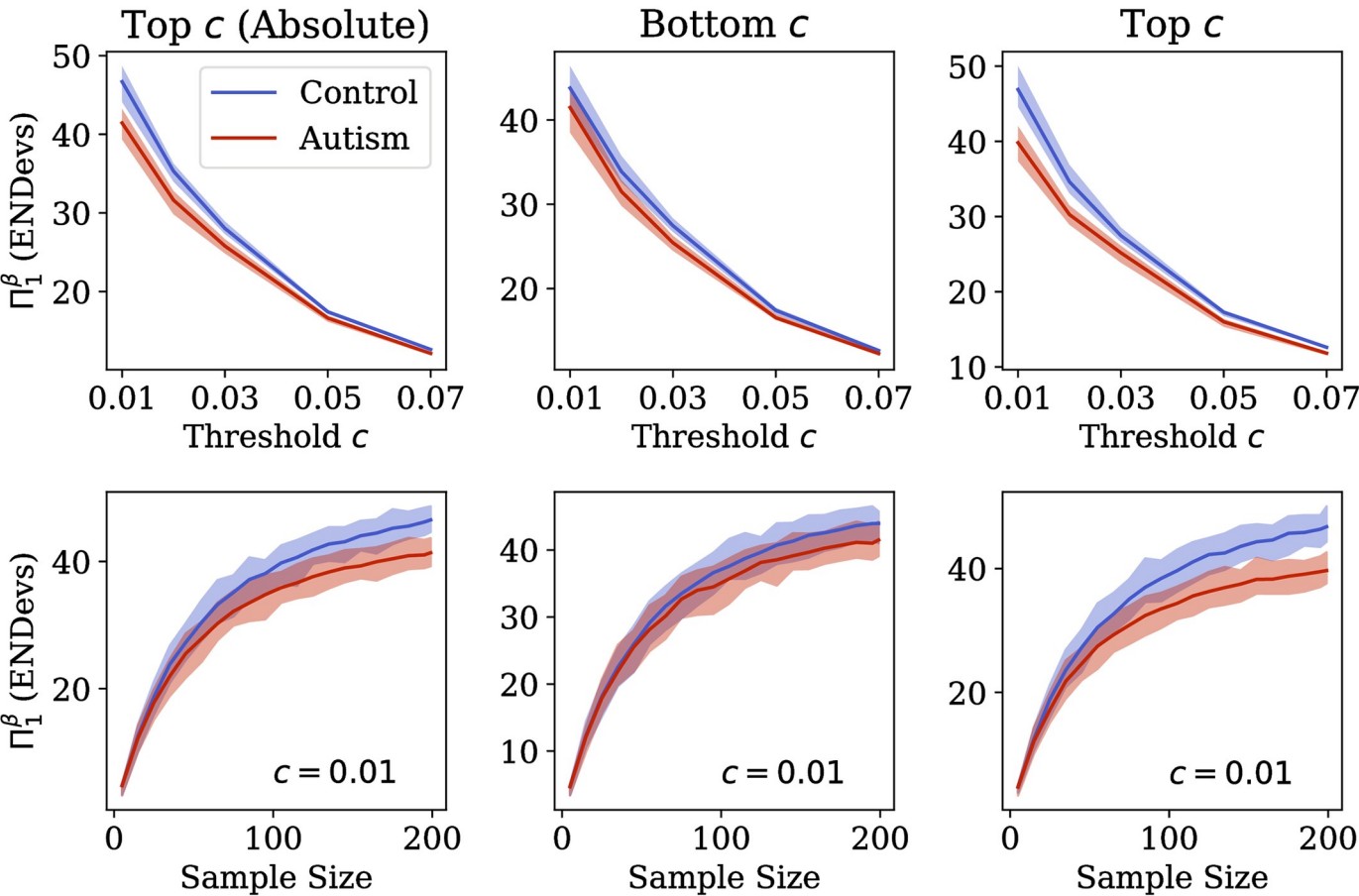

**Fig 5. Results of experiment on ABIDE dataset. Top Row**: The effective number of deviation patterns (ENDevs; y-axes) across thresholds $c$ for control (blue) and autism (red) subgroups. **Bottom Row**: Heterogeneity accumulation curves (ENDevs against sample size; at $c = 0.01$). The leftmost column shows results for the top $100c$% of absolute standardized deviance values, while centre and right columns show results for the lowest and highest $100c$% of standardized deviance values, respectively. Solid lines are means and ribbons are 95% confidence intervals.

## Discussion

This paper has identified a method of measuring heterogeneity in clinical cohorts under the normative modeling paradigm. Using Rényi heterogeneity, we quantified the effective number of distinct and equiprobable patterns of deviation from a normative distribution, guaranteeing

**Table 1. Tabular representation of experimental results with ABIDE dataset.**

| Threshold $c$ | Top 100%$c$ (Absolute) | | Bottom 100%$c$ | | Top 100%$c$ | |
|---|---|---|---|---|---|---|
| | Normative $\Pi_1^\beta$ | Clinical $\Pi_1^\beta$ | Normative $\Pi_1^\beta$ | Clinical $\Pi_1^\beta$ | Normative $\Pi_1^\beta$ | Clinical $\Pi_1^\beta$ |
| 0.01 | 46.4 [44.4,48.4] | 41.1 [38.8,43.4] | 43.9 [40.8,46.0] | 41.7 [38.0,44.2] | 46.8 [43.6,49.4] | 39.9 [37.8,42.1] |
| 0.02 | 35.5 [34.0,36.8] | 31.5 [30.2,32.6] | 33.9 [32.1,35.1] | 31.6 [30.4,32.9] | 34.6 [33.2,37.0] | 30.3 [29.1,31.7] |
| 0.03 | 28.1 [27.3,28.9] | 25.9 [25.2,26.9] | 27.5 [26.2,28.3] | 25.4 [24.8,26.7] | 27.5 [26.2,28.2] | 25.1 [24.0,26.1] |
| 0.05 | 17.4 [16.9,17.7] | 16.6 [16.2,17.0] | 17.4 [17.1,17.7] | 16.6 [16.2,16.9] | 17.3 [17.0,17.7] | 16.0 [15.6,16.5] |
| 0.07 | 12.6 [12.5,12.8] | 12.1 [11.9,12.3] | 12.7 [12.4,12.8] | 12.3 [12.1,12.5] | 12.6 [12.4,12.8] | 11.8 [11.6,12.1] |

The effective number of typical deviation profiles ($\Pi_1^\beta$) for control (Normative) and autism-diagnosed (Clinical) subjects across various extreme value thresholds (first column).

that heterogeneity is being measured in an axiomatically sound fashion consistent with other scientific disciplines [11, 15].

Our method exploits a potential synergy between normative modeling [1] and representational Rényi heterogeneity (RRH) [22]. The RRH theory was developed after previous work showed that there were no existing heterogeneity measures capable of application to the types of data used in modern psychiatric research [11, 34]. Furthermore, RRH was shown to generalize heterogeneity measures used across multiple disciplines, such as ecology [15, 18], economics [19], and statistical physics [24]. Representational Rényi heterogeneity involves measuring the size of a system's event space, which can be generated (along with a probability distribution over it) by a normative model. In the present study, this was the space of $c$-deviation patterns, where $c$ was an extreme value threshold. If this set of deviation patterns constitutes a relevant representation for the condition of scientific interest, then Rényi heterogeneity will provide an axiomatically sound measure of diversity in that set [11, 15, 16, 20, 26].

In all applications within our study, the ENDevs was lower for groups hypothesized to lie outside of some normative range. In the fully synthetic and MNIST experiments, samples drawn from outside the normative range showed smaller ENDevs than samples drawn from within the normative distribution. This behaviour was also shown in our experiment with the ABIDE data (Fig 5), albeit less markedly. This may be due to several factors. Namely, the heterogeneity accumulation curves did not show complete plateau, which suggests that the sample sizes are inadequate. It is also possible that diagnostic labels by which the clinical cohort was identified may define an inherently heterogeneous group, or the features/model may not suffice to capture the specific representation along which the diagnostic label is most homogeneous. Indeed, the ultimate goal of biological psychiatric research is to find feature representations—whether at genetic, molecular, structural, or other levels of analysis—that facilitate coherent definition of homogeneous clinical groups.

Representational Rényi heterogeneity is a general theory of heterogeneity that can be applied to any dataset, given a suitable probabilistic representation [22]. Since normative models can generate probability distributions over patterns of deviation from normal variation, RRH (here with units of ENDevs) can therefore be measured in any normative modeling study, although the normative modeling framework at present has been largely isolated to neuroimaging applications [1]. Future work should extend normative modeling, and consequently our heterogeneity measure, to genetic, connectomic, and behavioural data. In principle, one may also derive various other RRH-derived indices for the normative modeling approach. New measures could be identified by defining other representations of deviation from the normative range; so long as one can also specify a probability distribution over such a representation, Rényi heterogeneity will be applicable, and the heterogeneity index will inherit its well understood properties [15, 20, 22].

Future work must formally identify the conditions under which clinical cohorts will have smaller effective numbers of deviation patterns than cohorts drawn from the normative distribution. For example, it is trivial to show that if the normative distribution is degenerate, then the clinical cohorts will almost certainly have a larger number of deviation patterns. This scenario of degeneracy is unlikely to be observed in real-world scenarios in which one would consider normative modeling to be useful. However, a complete understanding of heterogeneity measurement and normative modeling requires that we understand all possible cases. Achieving this will require further formal analysis.

In conclusion, we have extended the popular and useful normative modeling approach with a heterogeneity measure identifying the ENDevs. Our measure is based on RRH and thus inherits its well understood properties, while ensuring that our definition of heterogeneity remains consistent with how heterogeneity is defined across other scientific disciplines

[11, 23, 24, 26]. Future work should (A) explore other useful Rényi heterogeneity measures derived from normative models, (B) apply normative modeling-based heterogeneity measures to other data modalities, and (C) provide further formal analyses of the behaviour of Rényi heterogeneity measures under normative models of different architectures.

## Supporting information

**S1 Fig. Rényi heterogeneity of Gaussian distributions.** Several univariate Gaussian distributions with mean 0 and standard deviations $\sigma_1, \sigma_2, \ldots, \sigma_5$ (probability densities shown in the left plot), and the corresponding Rényi heterogeneity values (right plot). On a continuous domain, such as that of a Gaussian distribution, the Rényi heterogeneity has units of "effective volume" (or length, or area, depending on the dimension).
(TIF)

**S2 Fig. Linear scaling of Rényi heterogeneity.** Demonstration of linear scaling of Rényi heterogeneity, in comparison to the variance and entropy on a unidimensional uniform distribution with domain size $u$ (i.e. the domain begins at the origin). Vertical and horizontal gridlines are set at 1.5 to illustrate that the Rényi heterogeneity increases by 50% when the domain size increases by 50%.
(TIF)

**S3 Fig. Variational loss under the MNIST data.** Variational loss for 10-folds of cross-validation within each digit class.
(TIF)

**S4 Fig. Deviation patterns in MNIST target cohorts.** Marginal distributions of most extremely deviant pixels (at the extreme value threshold $c = 0.01$) for MNIST digit classes. Marginalization was done over the images of the "clinical" digit class. The digits listed along rows are the "normative" classes. The digits listed along the columns are the "clinical" cohorts. For example, the image in the top-right corner (row 0, column 9) depicts the pattern by which images of "Nines" tend to deviate from a normative distribution of "Zeros," as modeled by a convolutional variational autoencoder [30, 31].
(TIF)

**S1 Table. Distribution of covariates across the normative (Control) and clinical (Autism) cohorts.** Covariates include site of origin (SITE), sex, handedness (HAND), age at scan (in years). Student's t-test was applied to evaluate difference in the mean of continuous variables between groups. The $\chi^2$ test was applied to evaluate differences in categorical variables between groups.
(CSV)

**S2 Table. Features from ABIDE dataset included after cross-validation.** These were features for which the normative model could explain more than 20 percent of variance across folds.
(XLSX)

**S1 File. Python code reproducing the synthetic data experiment.**
(IPYNB)

**S2 File. Python code reproducing the MNIST experiment.**
(IPYNB)

**S3 File. Python code reproducing the experiment on autism neuroimaging data.**
(IPYNB)

**S4 File. Demographic and clinical variables for the autism neuroimaging experiment.**
(CSV)

**S5 File. Neuroimaging variables for the autism neuroimaging experiment.**
(CSV)

## Author Contributions

**Conceptualization:** Abraham Nunes.

**Data curation:** Abraham Nunes.

**Formal analysis:** Abraham Nunes.

**Funding acquisition:** Martin Alda.

**Investigation:** Abraham Nunes.

**Methodology:** Abraham Nunes.

**Project administration:** Abraham Nunes.

**Resources:** Abraham Nunes.

**Software:** Abraham Nunes.

**Supervision:** Thomas Trappenberg, Martin Alda.

**Validation:** Abraham Nunes.

**Visualization:** Abraham Nunes.

**Writing – original draft:** Abraham Nunes.

**Writing – review & editing:** Thomas Trappenberg, Martin Alda.

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
