## [Decision Letter · Decision Letter 0]

29 Sep 2020

PONE-D-20-27446

Measuring Heterogeneity in Normative Models as the Effective Number of Deviation Patterns

PLOS ONE

Dear Dr. Nunes,

Thank you for submitting your manuscript to PLOS ONE. After careful consideration, we feel that it has merit but does not fully meet PLOS ONE’s publication criteria as it currently stands. Therefore, we invite you to submit a revised version of the manuscript that addresses the points raised during the review process.

ACADEMIC EDITOR:

The choice of synthetic and autism dataset was understood but the use handwritten dataset was not explained elaborately (in relation to this study). Have similar studies used these datasets? Is it possible to adapt datasets used by similar study with the proposed methodology?

We look forward to receiving your revised manuscript.

Kind regards,

Sudipta Roy

Academic Editor

PLOS ONE

Journal Requirements:

Additional Editor Comments (if provided):

Figures captions are too long. Good to describe in text in the main manuscript.

Reviewers' comments:

Reviewer's Responses to Questions

**Comments to the Author**

1. Is the manuscript technically sound, and do the data support the conclusions?

Reviewer #1: Yes

Reviewer #2: Yes

Reviewer #3: Yes

Reviewer #4: Yes

2. Has the statistical analysis been performed appropriately and rigorously? 

Reviewer #1: Yes

Reviewer #2: Yes

Reviewer #3: Yes

Reviewer #4: Yes

3. Have the authors made all data underlying the findings in their manuscript fully available?

Reviewer #1: No

Reviewer #2: No

Reviewer #3: Yes

Reviewer #4: Yes

4. Is the manuscript presented in an intelligible fashion and written in standard English?

Reviewer #1: Yes

Reviewer #2: No

Reviewer #3: Yes

Reviewer #4: Yes

5. Review Comments to the Author

Reviewer #1: 1 The first dataset needs more information about the type of data and explanation of data like second dataset.

2 Is probability distribution random?

3 “unlike variance and entropies, a 50% increase in heterogeneity corresponds to a 50% increase of variation in the system. - It needs explanation.

4 What is c-deviant?

5 The method generated data from this system for N = 50 subjects in a “normative cohort"- How you get the value N=50 as subjects?

6 Sampling rate is not specified.

7 Specify result of the method in a tabular form and there is a comparative study between your method and exiting methods.

Reviewer #2: Add motivation section.

Figures are also stretched.

Reviewer #3: I believe the paper is well written and clearly documents the findings of the research work undertaken. I am a bit concerned about the length of the Figure captions. The Figure captions are too long. Additionally, a typographic error in Line No. 178 needs to be corrected.

I do understand these are minor issues that can be easily corrected.

Given that all data is accessible, and all code has been duly provided as supplementary material, I suggest the work be accepted for publication with minor changes.

Reviewer #4: The authors have extended the normative modelling framework for measuring the amount of heterogeneity in clinical cohort. The authors used Representational Renyi Heterogeneity method that generalizes diversity measurement paradigms.

The paper is well written and with adequate statistics in support of the claims made by the authors. However, minor issues needs to be addressed by the authors to make the article more readable.

The introduction section should explain the problem at hand more elaborately.

The authors have not emphasized the necessity (novelty) of this work.

The review of past work has been covered in a sentence with a series of references; however, the readers may not be well aware of the subject. A paragraph towards the same should suffice.

The authors should consider explaining the images in text and reduce the lengthy explanations in figure captions.

The choice of synthetic and autism dataset was understood but the use handwritten dataset was not explained elaborately (in relation to this study). Have similar studies used these datasets? Is it possible to adapt datasets used by similar study with the proposed methodology.

Discussion section should include comparative study with other similar work (at least 2-3 articles that have approached the problem at hand).

6. PLOS authors have the option to publish the peer review history of their article (what does this mean?). If published, this will include your full peer review and any attached files.

Reviewer #1: No

Reviewer #2: No

Reviewer #3: No

Reviewer #4: No

---

## [Author Response · Author response to Decision Letter 0]

15 Oct 2020

Thank you for the insightful and helpful criticism, as well as the opportunity to revise our manuscript. We have appended all responses to reviewers in a letter attached to this submission.

Sincerely, 

Abraham Nunes, Martin Alda, and Thomas Trappenberg

---

## [Decision Letter · Decision Letter 1]

2 Nov 2020

Measuring Heterogeneity in Normative Models as the Effective Number of Deviation Patterns

PONE-D-20-27446R1

Dear Dr. Nunes,

We’re pleased to inform you that your manuscript has been judged scientifically suitable for publication and will be formally accepted for publication once it meets all outstanding technical requirements.

Kind regards,

Sudipta Roy

Academic Editor

PLOS ONE

Additional Editor Comments (optional):

The paper quantifies the heterogeneity in a cohort using normative modeling framework. The method is technically sound and well organized. The paper is now framed as good paper for publication.

Reviewers' comments:

Reviewer's Responses to Questions

**Comments to the Author**

1. If the authors have adequately addressed your comments raised in a previous round of review and you feel that this manuscript is now acceptable for publication, you may indicate that here to bypass the “Comments to the Author” section, enter your conflict of interest statement in the “Confidential to Editor” section, and submit your "Accept" recommendation.

Reviewer #1: All comments have been addressed

Reviewer #3: All comments have been addressed

Reviewer #4: All comments have been addressed

2. Is the manuscript technically sound, and do the data support the conclusions?

Reviewer #1: Yes

Reviewer #3: Yes

Reviewer #4: Yes

3. Has the statistical analysis been performed appropriately and rigorously? 

Reviewer #1: Yes

Reviewer #3: Yes

Reviewer #4: Yes

4. Have the authors made all data underlying the findings in their manuscript fully available?

Reviewer #1: Yes

Reviewer #3: Yes

Reviewer #4: Yes

5. Is the manuscript presented in an intelligible fashion and written in standard English?

Reviewer #1: Yes

Reviewer #3: Yes

Reviewer #4: Yes

6. Review Comments to the Author

Reviewer #1: The proposed approach is new one and it is proved also that it does not compromise with the accuracy of the method. Normative Modeling is the basic agenda of solving the problem. All corrections specified earlier are corrected.

Reviewer #3: (No Response)

Reviewer #4: The authors have responded well to all the comments and suggestion made by the reviewers. They have updated the paper with the suggestions made by the reviewers. The paper is better in all respect. Thanks to the authors.

7. PLOS authors have the option to publish the peer review history of their article (what does this mean?). If published, this will include your full peer review and any attached files.

Reviewer #1: No

Reviewer #3: No

Reviewer #4: No

---

## [Editor Report · Acceptance letter]

4 Nov 2020

PONE-D-20-27446R1 

Measuring Heterogeneity in Normative Models as the Effective Number of Deviation Patterns 

Dear Dr. Nunes:

I'm pleased to inform you that your manuscript has been deemed suitable for publication in PLOS ONE. Congratulations! Your manuscript is now with our production department. 

Kind regards, 

on behalf of

Dr. Sudipta Roy 

Academic Editor

PLOS ONE